# Heterotrophic Microbiota from the Oligotrophic Waters of Lake Vostok, Antarctica

**DOI:** 10.3390/ijerph19074025

**Published:** 2022-03-29

**Authors:** Ekaterina Y. Epova, Alexei B. Shevelev, Ramazan M. Akbayev, Yulia K. Biryukova, Marina V. Zylkova, Elena S. Bogdanova, Marina A. Guseva, Yaroslav Y. Tynio, Vladislav V. Egorov

**Affiliations:** 1Vavilov Institute of General Genetics, Russian Academy of Sciences, 119991 Moscow, Russia; cat-epova@yandex.ru (E.Y.E.); shevel_a@hotmail.com (A.B.S.); mary.zyl@mail.ru (M.V.Z.); ledera@yandex.ru (E.S.B.); marina.rom@mail.ru (M.A.G.); yytynio@mail.ru (Y.Y.T.); 2Skryabin Moscow State Academy of Veterinary Medicine and Biotechnology, 109472 Moscow, Russia; acbay@yandex.ru (R.M.A.); egoroffvlad1@gmail.com (V.V.E.)

**Keywords:** Antarctica, lake, fresh water, microbiota, 16S rDNA, chemoorganotrophs, *Microbacterium testaceum*, *Microbacterium trichothecenolyticum*, *Brevundimonas diminuta*, *Sphingomonas oligophenolica*, *Sphingomonas*, *Sphingobium limneticum*, *Dendryphion*, *Cladosporium fusiforme*

## Abstract

Lake Vostok is the deepest lake of Antarctica but has poor accessibility for study due to a thick glacial cover, however, water samples of this lake have become available for study just recently. Previously, only the microbiome of the ice cover samples was characterized. Here we report results of bacteriological seeding with subsequent identification of the heterotrophic microorganisms (bacteria and micellar fungi) present by 16S rDNA sequencing as well as results of a direct molecular study of the water microbiome. Surprisingly, the data obtained gave evidence of a predominant occurrence of common chemoorganotrophs that were rather psychrotolerant than psychrophilic. We isolated and described strains belonging to eight heterotrophic microbial species able to grow in a rich medium: six bacterial strains belonging to the species *Microbacterium testaceum* and *Microbacterium trichothecenolyticum*, *Brevundimonas diminuta*, *Sphingomonas oligophenolica*, *Sphingomonas* sp. and *Sphingobium limneticum*; and two fungal strains belonging to *Dendryphion* sp. and *Cladosporium fusiforme*. Direct study of 16S rDNA purified water samples confirmed the predominance of the *Brevundimonas*, *Microbacterium, Bradyrhizobium*, and *Bacillus* (*Bacillus cereus*) *genera*.

## 1. Introduction

Lake Vostok was discovered over the period of 1959 to 1964 by a method of seismic exploration [1]. In 1998, drilling of the ice cover with a heat drill started at the Russian Antarctica station Vostok (Figure 1a). In parallel, in 1961, [2] had proposed a theoretic justification of the ice melting under high-pressure conditions caused by a high mass of Antarctic glaciers.

Drilling of the 5G1 well at Vostok station began in 1989. In 1990, the 35th Antarctic Expedition reached a depth of 1250 m. In 1992, scientists switched to a smaller bore diameter, and in 1995, in the range of 2755–3350 m, the drilling method was changed from thermal to mechanical. In 1998, the International Scientific Committee on Antarctic Research made recommendations to Russia to suspend drilling at Vostok station to avoid chemical and microbial contamination of the lake waters. On 5 February 2012, the expedition reported that they had reached the surface of Lake Vostok at the depth 3769.3 m. In mid-December 2014, the expedition began their work at 3724 m and suspended it on 14 January 2015, only 4 m from the lake’s surface. On 24 January 2015, the expedition extracted 2 m cores from the mine, and on 25 January, starting from a depth of 3767.57 m, they raised two ice cores to the surface and reached the surface of Lake Vostok [3].

The first publication about the microbiota of Lake Vostok appeared in 1999 [4]. The authors found viable microorganisms in the ice core; their concentration was estimated as 10^2^–10^3^ colony-forming units (cfu)/mL. Later [5] carried out a metagenomic analysis of DNA isolated from the Vostok ice core at a depth of 3563–3585 m (V5 sample) and 3606–3621 m (V6 sample). They found the predominant occurrence of nitrogen-fixing bacteria (*Azospirillum*, *Azotobacter*, *Bacillus*, *Burkholderia*, *Frankia*, *Klebsiella*, *Rhizobium*, *Rhodobacter*, *Rhodopseudomonas*, *Sinorhizobium*, *Methylococcus*, *Nitrobacter*, *Nitrococcus*, *Notrosococcus*, and *Nitrosomonas*). Species important in other parts of the nitrogen cycle were within the genera *Alcaligenes*, *Bacillus*, *Clostridium*, *Micrococcus*, *Paracoccus*, *Proteus*, *Pseudomonas*, *Streptomyces*, and *Thiobacter*.

Classification of the microorganisms by carbon fixation indicated that most of the organisms utilize either the reductive tricarboxylic acid (rTCA) cycle (Alphaproteobacteria, Betaproteobacteria, Gammaproteobacteria, and Chlorobi) or the reductive pentose phosphate (rPP or Calvin–Benson) cycle (Archaeplastida, Chromalveolates, Cyanobacteria, Alphaproteobacteria, Betaproteobacteria, and Gammaproteobacteria). Based on the frequency of gene sequences, the most common mode of CO_2_ fixation was the rTCA cycle, while the rPP cycle was the second most common. The two Archaea in V5 may fix carbon via the reductive acetyl-CoA (rACA) pathway. A large number and diversity of sequences from phototrophs were present in the accretion ice, including 228 cyanobacterial, 11 algal, 12 chromalveolate, and other unique sequences. About 6% of the unique sequences were closest to eukaryotes (221 from V5, 87 of which have available sequence identities and 29 from V6, 24 of which available sequence identities), and diverse taxonomic groups were represented. Most of the sequences were most similar to those from Fungi (91 sequences in V5 and 22 in V6), including one small subunit rRNA sequence that was 99% similar to a marine fungus sequence that had been recovered from a deep-sea thermal vent. Several sequences from species of Animalia were found, including 21 sequences closest to those from arthropods (16 in V5 and 5 in V6), many of which are predatory or parasitic, including sequences closest to species of *Daphnia* (planktonic crustaceans; 98% identity) and others.

It was suggested that most DNA sequences from the water of Lake Vostok reported earlier should be considered as contaminants that appeared during the course of the drilling [6]. They emphasized finding the chemolithoautotrophic thermophile bacteria *Hydrogenophilus thermoluteolus* (Betaproteobacteria), an actinobacterium related to *Ilumatobacter fluminis* and unclassified bacterium AF532061, which should be considered a true indigenous species of Lake Vostok. They also suggested that this ecosystem did not contain Archaea.

Analysis of the microbiota in Lake Vostok should consider the available data regarding the microbiota of other Antarctic lakes, although their geophysical conditions differ substantially; the most studied of these is Lake Fryxell [7]. Other examples are Lakes Joyce [8], Vida [9], Stain house [10], and Whillans [11]. In our opinion, the conditions closest to Lake Vostok are found in Lake Vanda [12]. The waters of Lake Vanda range from fresh and cold (~4 °C) just below the ice cover to hypersaline and warm (~24 °C) just above the sediments. At depths of 50–55 m, the Lake Vanda waters are approximately 10–12 °C with a pH of 6.5–7.0. The level of dissolved organic carbon (DOC) in Lake Vanda is quite low (<0.2 mg/L) down to about 60 m, and salinity is nearly undetectable to a depth of 55 m. In addition, the lake is supersaturated with dissolved O_2_ to a depth of about 65 m, beyond which salinity, temperature, and sulphide increase to the sediments [13]. Six strains of psychrotolerant, heterotrophic bacteria were isolated from lake water samples from a depth of 50 or 55 m. Phylogenetic analyses showed that the Lake Vanda strains belonged to families Nocardiaceae, Caulobacteraceae, Sphingomonadaceae, and Bradyrhizobiaceae. All lake Vanda strains were able to grow at temperatures near or below 0 °C, but optimal growth occurred in the range from 18 to 24 °C. Some strains showed halotolerance, but no strains required a high NaCl concentration for growth.

Taking into account data about the predominant occurrence of mesophilic, heterotrophic strains in the fresh-water layers of Lake Vanda and the low share of extremophiles and anaerobic species, we pursued priority characterization of the species able to grow on complete organic media in the presence of oxygen under mesophilic conditions.

Taking into account that the microbiome of the Lake Vostok water was never previously characterized, the goal of the study included the discovery of the massive components of the water microbial community, including both culturable and unculturable components. Isolation of alive isolates was considered as a priority since it allowed to characterize the physiological properties of the microbiome members. PCR cloning of 16S rDNA isolated directly from the raw samples collected from the Lake Vostok were compared to the data of microbiological seeding. Elective media allowing isolation of saprotrophic fungi was used along with elective media for the cultivation of the chemotrophic bacteria. A question about natural sources of carbon and energy for the chemoheterotrophic microbiome of Lake Vostok was addressed.

## 2. Materials and Methods

### 2.1. Sample Collection and Analysis, Enrichment and Isolation

The samples of Vostok water were collected from the 5G borehole having GPS co-ordinates 72.28° S 106.48° E [6] by using a heat drill head fitted with joined sections of a 10-inch (25.4 cm) steel auger to create the sampling hole in the 3539 m lake ice cover (Figure 1b). Water samples were collected in a 5 L Niskin sampling bottle and transferred to completely filled polypropylene tubes with screw caps (15 mL each) that had been previously rinsed with a 10% HCl solution and rinsed 10 times with distilled water. All samples were kept in a refrigerator (+4 °C) in the dark until used for seeding (~1 month).

Primary cultures were established by aseptically spreading 0.5 mL of water from Lake Vostok onto Petri dishes containing 1× LB (Bacto Peptone 10 g/L, yeast extract 5 g/L and NaCl 10 g/L), 0.5× LB or 0.1× LB; Pseudomonas Agar (Merck cat. #10230, Na-L-glutamate 10 g/L, water-soluble starch 20 g/L, KH_2_PO_4_ 2 g/L, MgSO_4_ 0.5 g/L, phenol red 0.36 g/L, agar 12 g/L, pH 7.2–7.4); Rhizobium Bean Agar (white bean extract 100 mL/L, K_2_HPO_4_ 0.5 g/L, sucrose 10 g/L, agar 20 g/L, pH 7.2–7.4); Rhizobium mannitol soil agar (fresh garden soil extract 200 mL/L, yeast extract 1 g/L, mannitol 10 g/L, agar 15 g/L); nitrate agar (Bacto beef extract [Difco] 3 g/L, Bacto peptone 5 g/L, KNO_3_ 1 g/L, agar 12 g/L); Photobacterium Agar (Difco cat. #0631-02, Bacto tryptone 5 g/L, yeast extract 2.5 g/L, NaCl 30 g/L, NH_4_Cl 0.3 g/L, MgSO_4_ 0.3 g/L, FeCl_2_ 0.01 g/L, CACO_3_ 1 g/L, KH_2_PO_4_ 3.0 g/L, Na glycerophosphate 23.5 g/L, agar 15 g/L, pH 7.0); or Staphylococcus agar (aminopeptide 13.1 g/L, extract of feeding yeast 5 g/L, NaCl 79 g/L, Na_2_HPO_4_ 0.5 g/L, agar 15 g/L). Incubations took place in the dark at +4 °C and at room temperature (~20 °C) for 7 days. The plates were examined daily. Appeared colonies selected for isolation were purified by at least three successive passages at the same medium which was used for primary plating. Isolated strains were suspended in LB broth, lyophilized, and kept in a refrigerator at +4 °C.

### 2.2. Morphology

Morphological examinations were carried out after Gram staining (×80, oil immersion field) [14] using a LOMO light microscope. Motility was determined by microscopic observations and from assessing stab inoculations into semi-solid (0.35% agar) sulfide-indole-motility (SIM) medium (Becton, Dickinson and Company, Sparks, MD, USA) [15].

### 2.3. Molecular Study of Uncultured Microbiota

Nucleic acid extraction from a 1 mL of a Lake Vostok water sample was performed using the MinElute Virus Spin Kit (Qiagen, Germantown, MD, USA) and eluted in 150 µL of AVE buffer. The eluted nucleic acid was further concentrated by precipitating overnight at −20 °C with 100 mM sodium acetate in 80% ethanol. They were then pelleted by centrifugation at 16,000× *g* for 15 min, washed with cold 80% ethanol and centrifuged at 16,000× *g* for 5 min. They were dried under a vacuum and then re-suspended in 30 µL TE buffer.

PCR cloning of the 16S rDNA fragment from the bacterial strains and the ribosomal ITS fragment from fungal strains was carried out with primers specified in Table 1.

Q5^®^ Hot Start High-Fidelity DNA Polymerase (New England Biolabs, Ipswich, MA, USA) with a buffer supplemented by the manufacturer was used for PCR. The PCR products were examined by electrophoresis in a 0.8% agarose gel stained with ethidium bromide and compared to a 1 kb DNA Ladder (Thermo Fisher Scientific, Whaltham, MA, USA). PCR products were purified without extraction from the agarose gel using the GeneJET Gel Extraction Kit (Thermo Fisher Scientific) and used for cloning to CloneJET PCR Cloning Kit (Thermo Fisher Scientific). The ligation mixture was used for transformation of *Escherichia coli* TG1 strain (supE, thi-1, Δ(lac-proAB) Δ(mcrB-hsdSM)5(rK-mK-) [F’ traD36 proAB lacI^q^ZΔM15]) and plated onto LB agar indicator medium supplemented with 2 mM isopropyl thiogalactoside and 0.02% X-gal.

Plasmid DNA was purified from 30 white colonies of the bacterial 16S rDNA library and 30 white colonies of the cyanobacteria library by using the GeneJET Plasmid Miniprep Kit (Thermo Fisher Scientific). Each plasmid DNA was subjected to digestion with BglII and analyzed by electrophoresis in a 0.8% agarose gel stained with ethidium bromide and compared to a 1 kb DNA Ladder (Thermo Fisher Scientific). The clones bearing a ~1.5 kb insertion were subjected to Sanger sequencing carried out by Eurogen LLC. (Moscow, Russia).

### 2.4. Molecular Phylogenetic Analyses

Genomic DNA from bacterial strains was isolated from fresh liquid cultures. Genomic DNA from fungal strains was isolated from mycelium that appeared on the agar plates. The cells were lysed by vortexing with beads of borosilicate glass (diameter 0.1 mm) in a 1:1 (*v*/*v*) water:phenol biphasic system. After three extractions of the water fraction with phenol:chloroform, nucleic acids were precipitated with 70% isopropanol.

PCR cloning of the 16S rDNA fragment from the bacterial isolates and the ribosomal ITS fragment from isolated fungal strains was carried out with primers specified in the previous section; Molecular Study of Uncultured Microbiota. The length of the PCR product was determined by electrophoresis in a 0.8% agarose gel stained with ethidium bromide and compared to a 1 kb DNA Ladder (Thermo Fisher Scientific). Each DNA fragment was purified by extraction from the agarose gel using the GeneJET Gel Extraction Kit (Thermo Fisher Scientific).

The obtained sequences of each PCR product were assembled and proofed manually by using the free SNAP gene software. Phylogenetic relationships of the newly determined sequences were determined using BLAST [20] and RDP [21]. MEGA5 was used to create maximum-likelihood phylogenetic trees with 1000 bootstrap replications using the Tamura–Nei distance correction model and the nearest neighbor interchange (NNI) heuristic method [22].

## 3. Results

### 3.1. Isolating Pure Cultures from the Microbiota of Lake Vostok

Characterization of the microbiota of Lake Vostok started from culturing raw samples collected from the lake on plates with complete media with varying concentrations of peptone, yeast extract, and NaCl. This decision was made based on the assumption that heterotrophs would likely be dominant in this reservoir, similarly to the upper layers of Lake Vanda [12]. Comparison of the plates with 1×, 0.5× and 0.1× LB agar incubated at +4 °C and at room temperature revealed no obvious differences in the appearance of the colonies. The only difference was the slower growth of the colonies as the nutrient content and temperature decreased. For example, 1 mL specimen of the water from Lake Vostok rendered ~65 big colonies (~30 brilliant orange and yellow colonies with smooth edges; ~25 white or cream-colored colonies with festoon edges; and ~10 brilliant white colonies with smooth edges) and ~300 small colonies with a vague color and morphology when plated onto 1× LB agar. The number of colonies did not depend on LB concentration, but the growth got slower along with the dilution of the substrate, and the smaller colonies were more visible nearby bigger ones.

A typical view of the colonies that appeared at 1× LB agar is shown in Figure 2.

This sample rendered ~1–2 white and black colonies with signs of micellar growth. Each type of colony was brought to a state of pure culture by three or more successive platings, which were carried out at room temperature. The parental colonies for obtaining pure cultures were taken from each of the six types of primary plates. This approach allowed us to achieve sustainable growth in all cases except colonies formed with black mycelium, which could be sustainably re-plated only if grown at +4 °C. At room temperature, the microorganism grew rapidly but then died fast.

Plating the raw water samples onto specific media (Pseudomonas Agar) allowed isolating additional groups of bacteria forming brownish transparent colonies and yellow colonies with or without a halo. Similar colonies appeared on Rhizobium Bean Agar.

Small-subunit rRNA gene sequences from bacterial and fungal strains of Lake Vostok isolated in this study were deposited into GenBank under the accession numbers shown in Table 2. Lyophilized cultures were deposited in the All-Russian Collection of Industrial Microorganisms. They are available upon request under the accession numbers in Table 2.

About 50% of the pigmented (orange and yellow) colonies that appeared in the primary cultures on LB agar were segregated into pigmented colonies with smooth edges (parental type) and white or cream-colored colonies with festoon edges when re-plated. Other pigmented colonies looked homogenous. The segregants of the initial pigmented colonies were re-plated 3–4 times in 1× LB medium; there was a clear difference in their stability. The pigmented segregants grew sustainably and did not show signs of non-homogeneity, whereas the white or cream-colored colonies stopped growing at the first or second passage.

The microscopic analysis revealed that both pigmented and white colonies with smooth edges were formed by motionless cocci, whereas the colonies with the festoon edges were composed of highly motile rods. As expected, the mixed pigmented colonies able to segregate into pigmented/smooth edge and creamy/festoon edge types were composed of motionless cocci and motile rods.

Isolates of small parental colonies of a vague morphology were always unambiguously allocated into one of three above-mentioned types when cultured for a longer time (3–4 days at room temperature): (1) motionless cocci forming orange and yellow colonies with a smooth edge; (2) motionless cocci forming white colonies with a smooth edge; (3) motile rods forming white or cream-colored colonies with a festoon edge. Molecular characterization of the taxonomic lineages was carried out by using at least 30 colonies of each type. It demonstrated that each type of the colony and cell morphology was represented by a single species: type 1 was *Microbacterium trichotecenolyticum* (Figure 3a), type 2 was *Microbacterium testaceum*, and type 3 was *Brevundimonas diminuta* (Figure 3b). The poorly re-plated colonies that segregated from the pigmented parental colonies morphologically belonged to type 3. Molecular examination confirmed that they also comprised strains of the species *B. diminuta*.

Molecular characterization of the mycelium-forming colonies allowed allocating them to fungi (Ascomycetes) and Actinobacteria. The most numerous black colonies able to grow sustainably at +4 °C only were identified as *Dendryphion* sp. (Pezizomycotina; leotiomyceta; dothideomyceta; Dothideomycetes; Pleosporomycetidae; Pleosporales; Torulaceae). The white colonies with a dense mycelium were *Cladosporium fusiforme* (Ascomycota; saccharomyceta; Pezizomycotina; leotiomyceta; dothideomyceta; Dothideomycetes; Dothideomycetidae; Cladosporiales; Cladosporiaceae). The colonies forming an openwork air mycelium were identified as *Corynebacterium pseudotuberculosis* (Actinobacteria; Actinobacteria; Corynebacteriales; Corynebacteriaceae).

Colonies that appeared after the primary plating on Pseudomonas Agar and Rhizobium Bean Agar did not grow on LB agar and exhibited only a slow growth on the specific media. On the basis of 16S rDNA sequencing, the most abundant brownish, transparent, highly mucous colonies were classified as *Sphingomonas* species; more rare yellow colonies without a halo were recognised as *Sphingomonas oligophenolica*; and the minor yellow colonies with a halo were *Sphingobium limneticum*. Strains belonging to *Brevundimonas* species did not appear on Pseudomonas Agar and Rhizobium Bean Agar, and peculiar orange *M. trichotecenolyticum* totaled only about 5% of colonies on Rhizobium Bean Agar (this diagnosis was confirmed by 16S rDNA sequencing).

No colonies appeared on Staphylococcus agar.

### 3.2. Characterization of Non-Culturable Portion of the Microbiota of Lake Vostok

Therefore, we have taken steps to verify the presence of the most abundant groups of organisms found in Antarctic lakes alongside the found Alphaproteobacteria, Actinobacteria, and Ascomycota. Further findings led to an expectation that the massive presence of active bacteria belonging to the family Bradyrhizobiaceae of the class Alphaproteobacteria [12]. We employed standard primers specific to the conserved region of 16S rDNA of Eubacteria: 8F and 1492R [17]. These primers were used for PCR amplification of DNA purified directly from the water of Lake Vostok. The mixed PCR products were separated by cloning in the plasmid vector. The size of the analyzed library was 30 clones. Identification of taxonomic lineages found by comparison of the sequences with the NCBI database is shown in Table 3.

## 4. Discussion

Viable microorganisms in Lake Vostok were first found in the ice core in 1999 at an estimated concentration of 10^2^–10^3^ colony-forming units (cfu)/mL. After reaching the surface of the lake at the depth 3769.3 m, later studies in 2012 carried out a metagenomic analysis of DNA isolated from the Vostok ice core [5]. It has been suggested that most DNA sequences from the water of Lake Vostok reported earlier should be considered as contaminants that appeared during the course of the drilling [6]. We report data about the investigation of the water samples independently picked up from Lake Vostok in 2018. We investigated the heterotrophic part of the unique ecosystem by isolating pure microbial cultures from samples taken in 2019. We isolated and described strains belonging to eight heterotrophic microbial species able to grow in a rich peptone-based medium at 20 °C: six bacterial strains belonging to the species *Microbacterium testaceum* and *Microbacterium trichothecenolyticum* (Actinobacteria), *Brevundimonas diminuta* (Alphaproteobacteria), *Sphingomonas oligophenolica* and *Sphingomonas* sp. (Alphaproteobacteria) and *Sphingobium limneticum* (Alphaproteobacteria); and two fungal strains belonging to *Dendryphion* sp. and *Cladosporium fusiforme* (Ascomycota). Direct PCR cloning of 16S rDNA from the water samples with 8F and 1492R confirmed the predominance of the *Brevundimonas*, *Microbacterium*, *Bradyrhizobium*, and *Bacillus* (*Bacillus cereus*) genera. The microbiota in Lake Vostok—mesophilic, psychrotolerant heterotrophs and the absence or low share of extremophiles and chemolithotrophs—resembles the one found in Lake Vanda (Western Antarctica) [12], but not the ones reported for Lakes Fryxell [7], Joyce [8], Whillance [11], Stain house [10] and Vida [9].

Two previous studies had focused on the molecular characterization of Lake Vostok [5,6]. The authors revealed a great variety of genome fragments belonging to different kingdoms of living organisms, for example, Eubacteria, Archaea, Fungi, Viridiplantae (chloroplasts), and even Metazoa (family Copepoda). However, interpretation of these data should take into account that a low temperature, absence of solar insolation, and contact with global biota may favor conservation of DNA from long-extinct organisms that may be mixed together with genomic fragments of the actual living organisms in the course of the treatment of the water samples. Moreover, the ice cover of Lake Vostok was examined, not the water [5]. A work by Bulat, 2016 is a review, which contains no experimental data about molecular studies of the microbiota [6]. On the other hand, data about the predominant occurrence of mesophilic/psychrotolerant cultivable heterotrophs in the water of Lake Vanda are in good agreement with our observations [12].

The performed analysis of the microbiota in the water of Lake Vostok demonstrated obvious similarities in its composition with the microbiota of Lake Vanda [12]. Similar to the upper layers of Lake Vanda, Lake Vostok is highly oligotrophic and has low salinity. Both lakes have a maximal ice core that protects the water from any traces of solar insolation. Like [12], we used conventional microbiological methods to study heterotrophs in the microbiota and confirmed the suggestion that they are represented by psychrotolerant rather than psychrophilic bacterial strains belonging to the classes Actinobacteria (genus *Rhodococcus* in Lake Vanda versus genus *Microbacterium* in Lake Vostok) and Alphaproteobacteria (genera *Brevundimonas* and *Bradyrhizobium* in both reservoirs). Strains of the genus *Bradyrhizobium* were not isolated as viable cultures apparently due to their slow growth on rich media compared with *Microbacterium* and *Brevundimonas*. However, based on molecular analysis of non-cultured samples, we hypothesise that strains of the genus *Bradyrhizobium* comprise the predominant share of the microbial community (43.3% *Bradyrhizobium* spp., 13.3% *Bradyrhizobium japonicum*, and 6.7% *Bradyrhizobium diazoefficiens*, as derived from representation in the library; Table 3), whereas strains of the genus *Brevundimonas* occupy the second place (10.0%; Table 3). The high frequency of mixed colonies including *M. trichothecenolyticum* and *B. diminuta* where *M. trichothecenolyticum* only conserves the ability of autonomous growth suggests the existence of trophic chains among the Lake Vostok microbiota, where strains of the genus *Brevundimonas* depend on metabolites of *M. trichothecenolyticum*. On the other hand, there are strains of *B. diminuta* that do not depend on presence of *M. trichothecenolyticum* in their environment. Of note, *B. diminuta* obviously dominating in the culturable portion of the Lake Vostok microbiota was not found among the predominant species of the non-cultured microbiota. The genus *Microbacterium* was not represented in the non-cultured library of 16S rDNA whereas *Bacillus cereus* and *Pelomonas* spp. occupying 3.3% of the 16S rDNA library were absent among the isolated pure cultures.

Analysis of the origin of the relatives of the found strains described elsewhere has shown that all three types of the isolated bacteria [23,24,25,26,27,28,29] (*M. trichothecenolyticum*, *M. testaceum* and *B. diminuta*) and both fungi [30] (*Dendryphion* sp. and *C. fusiforme*) are close to strains that form associations with algae (*Chlorella*) or plants (Table 2). All strains of *Bradyrhizobium* found in the non-cultured 16S rDNA libraries have close relatives associated with plants. Only minor components of the non-cultured library—*Brevundimonas* spp., *B. cereus*, and *Pelomonas* spp.—are related to strains found in fresh water or Antarctic seawater. Strains of *Sphingomonas* sp. and *S. oligophenolica* are inhabitants of soils, including antimonite mining tails containing arsenate. *S. limneticum* is a species attributed to seawater and freshwater in moderate latitudes, for example, lignin-containing sediment in Kagoshima Bay, Japan, and a freshwater green alga, *Paulinella chromatophora* (Table 1).

This observation led us to an assumption that the microbial community of Lake Vostok should be trophically associated with phototrophs. As reported by [5,6], cyanobacteria were the most abundant representatives of the phototrophs in Lake Vostok. Therefore, we searched for this group of microorganisms by using specific primers to 16S rDNA (see Materials and Methods). However, no specific PCR products were found in the total DNA isolated from the Lake Vostok water sample.

The domination of mesophilic heterotrophs (both bacterial and fungal) exhibiting signs of association with phototrophs raises a hypothesis about the phototrophic nature of primary biogenic production in Lake Vostok and presumably in other Antarctic lakes. This hypothesis is not in agreement with the fact that a thick ice cover completely absorbs the solar light and prevents its penetration into the water of the Antarctic lakes. We propose that there is a non-solar source of light in Lake Vostok and presumably other Antarctic lakes that explains this contradiction.

## 5. Conclusions

The fresh water of Lake Vostok contains 10^2^–10^3^ cfu heterotrophic bacteria mostly represented by species *Microbacterium testaceum*, *Microbacterium trichothecenolyticum*, and *Brevundimonas diminuta*. Species Sphingomonas sp., *Sphingomonas oligophenolica*, and *Sphingobium limneticum* are found as minor components of culturable heterotrophic aerobic and aerotolerant microbiome. Heterotrophic Ascomycetes fungi, e.g., *Dendryphion* sp. and *Cladosporium fusiforme* are found in the lake in concentration below 1 c/f/u/per 1 mL water. All mentioned species except *Dendryphion* sp. are moderate thermal conditions of growth to temperature ~4 °C. Data of direct PCR cloning of the Lake Vostok water samples demonstrate the predominance of species *Bradyrhizobium* spp., *Bradyrhizobium japonicum* and *Bradyrhizobium diazoefficiens* (major components) and *Brevundimonas* spp., *Bacillus cereus* and *Pelomonas* spp. (minor components).

Taken together, our data make addressing a question of the ecological structure of the microbial community of the surface part of the water of Lake Vostok. No sufficient organic material was found there for providing appropriate energy and carbon source for the found heterotrophic microorganisms. Therefore, one should hypothesize a presence of a major source of organic material input in this reservoir.

## Figures and Tables

**Figure 1 ijerph-19-04025-f001:**
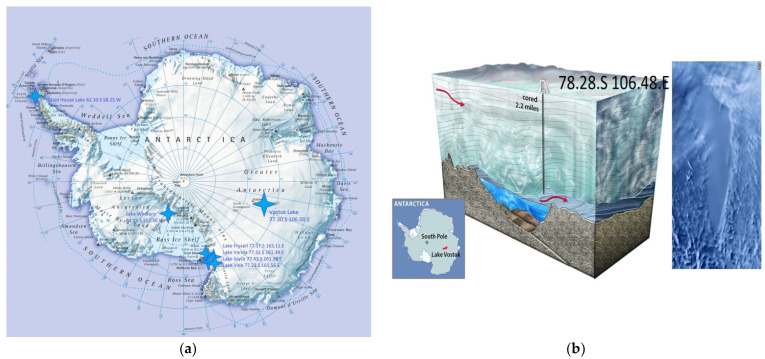
Location of the most studied lakes in Antarctica (**a**) and 5G borehole in Lake Vostok (**b**).

**Figure 2 ijerph-19-04025-f002:**
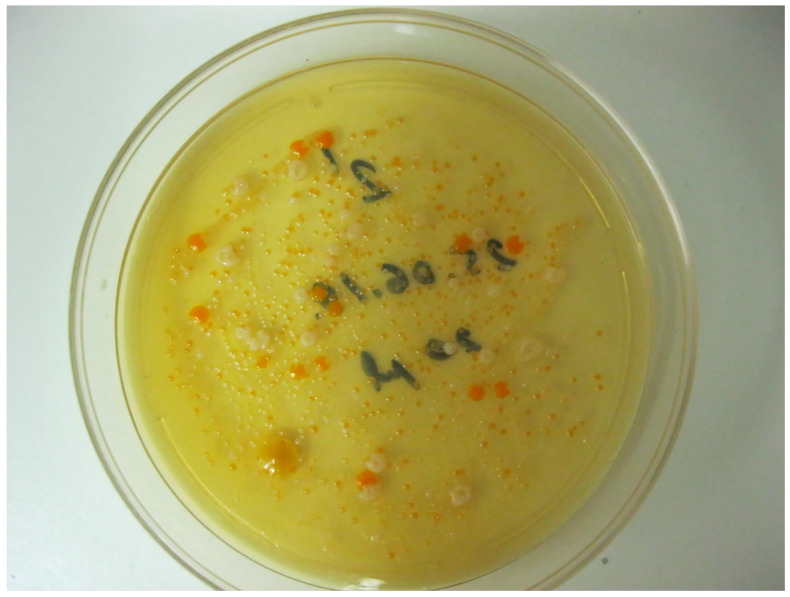
Typical view of a Petri dish with 1× LB agar after plating 0.5 mL of a water sample and incubation for 3 days at room temperature.

**Figure 3 ijerph-19-04025-f003:**
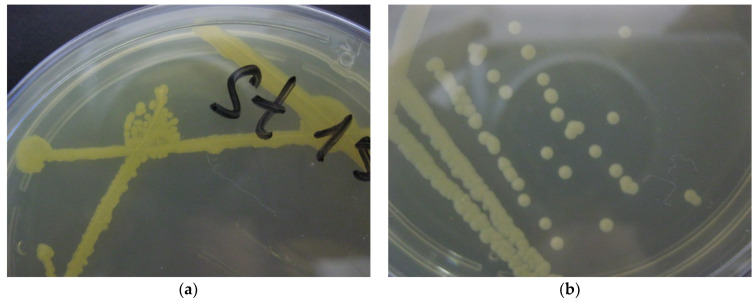
Typical view of a Petri dish with 1× LB agar with plated *Microbacterium trichotecenolyticum*—type 1 (**a**) and *Brevundimonas diminuta*—type 3 (**b**).

**Table 1 ijerph-19-04025-t001:** Primers used PCR cloning of the 16S rDNA fragment from the bacterial strains and the ribosomal ITS fragment from fungal strains.

Group of Microorganisms	Primer Name	Primer Sequence	Reference
Eubacteria 16S	8F	5′-AGAGTTTGATCCTGGCTCAG-3′	[16,17]
	1492R	5′-TACCTTGTTACGACTT-3′	
Cyanobacteria 16S	8F	5′-AGAGTTTGATCCTGGCTCAG-3′	[18]
	1480R	5′-AGTCCTACCTTAGGCATCCCCCTCC-3′	
Ascomycetes ITS	ITS4Asc	5′-CGTTACTRRGGCAATCCCTGTTG-3′	[19]
	ITS5	5′-GGAAGTAAAAGTCGTAACAAGG-3′	
Oomycetes ITS	ITS4Oo	5′-ATAGACTACAATTCGCC-3′	
	ITS5	5′-GGAAGTAAAAGTCGTAACAAGG-3′	
Zygomycetes ITS	ITS4Zygo	5′-AAAACGTWTCTTCAAA-3′	
	ITS5	5′-GGAAGTAAAAGTCGTAACAAGG-3′	

**Table 2 ijerph-19-04025-t002:** Pure cultures of heterotrophs isolated from the water of Lake Vostok by plating on LB agar.

Name of Culture	NCBI GenBank Acc. # of 16S/18S rDNA Fragment ^1^	Registration No. in All-Russian Collection of Industrial Microorganisms	Appearance of Colonies	Cell Morphology/Temperature Requirements	Description of the Nearest Relative, Described Elsewhere
*Microbacterium trichotecenolyticum* St15	MT192546.1	VKPM B-13662	Brilliant, orange or yellow colonies with smooth edge	Motionless cocci	[23,24]
*Microbacterium testaceum* St11	MT192545.1	VKPM B-13661	Brilliant, white with smooth edge	Motionless cocci	[25]
*Brevundimonas diminuta* St10	MT192544.1	VKPM B-13660	White/creamy with festoon edge	Motile rods	[26]
*Sphingomonas* sp. St1	MW316031.1	NA	Brownish highly mucous transparent colonies	Motionless rods	[27], NCBI Acc. #AB974275.1
*Sphingomonas oligophenolica*	MW316054.1	NA	Yellow colonies	Motionless rods	Lee D., Lee Y., Lee J.C., Yang K.S., Kim C.S., Oh D.,-J., and Jung Y.,-H. Microbial diversity from soil in Jeju Island, *Sphingomonas mali* strain JBRI-MO-0011, NCBI Acc. #MK302226
*Sphingobium limneticum* St2	MW316043	NA	Yellow colonies with halo	Motionless rods	[28,29]
*Dendryphion* sp.	MW341486.1 MW341467.1	NA	Black dense mycelium	Psychrophilic	*Dendryphion europaeum* isolate E19/16-9; antagonism between the fungus *Eutypella parasitica* and selected fungal species in the wood of dead branches of sycamore maple (*Acer pseudoplatanus*); *Torula herbarum* strain CBS 220.69
*Cladosporium fusiforme*	MW341490.1 MW341491.1	NA	White dense mycelium	Mesophilic	*Cladosporium fusiforme* NCBI Acc. # CBS 119414, [30]

^1^ https://www.ncbi.nlm.nih.gov/nuccore/, accessed on 20 February 2022.

**Table 3 ijerph-19-04025-t003:** The taxonomic lineage of non-cultured eubacteria from the water of Lake Vostok.

Species Name	Number of Plasmid Clones in the Library	Closest Formerly Reported Sequence		Closest Formerly Reported Cultured Isolate	
		Description	% of Identity	Description	% of Identity
*Bradyrhizobium* spp.	13	*Bradyrhizobium* sp. (NCBI Acc. # ^1^ MK848653), source: ferruginous soil of eastern Amazonia	98.8	*Bradyrhizobium* sp. strain UFLA04-893 (NCBI Acc. #MK848653), source: ferruginous soil of eastern Amazonia	96.7
*Bradyrhizobium japonicum*	4	*Bradyrhizobium* sp. CCBAU 23332 (NCBI Acc. #HQ428040), source: nodules of *Glycine max*, China	99.2	*Bradyrhizobium japonicum* strain MSDJ 5697 (NCBI Acc. #AF363135)	98.3
*Bradyrhizobium diazoefficiens*	2	Uncultured proteobacterium clone Amb_16S (NCBI Acc. #EF018606), source: soil, Argentina	100	*Bradyrhizobium diazoefficiens* strain S04K (NCBI Acc. #EF018606), source trembling aspen rhizosphere, New York, USA	99.1
*Brevundimonas* spp.	3	Uncultured bacterium clone EMIRGE_OTU_s3t2d_1061 (NCBI Acc. #JX222695), source: subsurface aquifer sediment, Colorado, USA	99.3	*Brevundimonas viscosa* strain 19SA03-R-5 (NCBI Acc. #MN540834), source: freshwater, South Korea	98.8
*Bacillus cereus*	1	Uncultured bacterium clone OTU103 (NCBI Acc. #KP975359), source: surface seawaters of Drake Passage near the Chinese Antarctic station	99.3	*Bacillus cereus* (NCBI Acc. #LN890165), rock surface, China	97.6
*Pelomonas* spp.	1	Uncultured bacterium clone OTU52 (NCBI Acc. #KP975308), source: surface seawaters of Drake Passage near the Chinese Antarctic station	100	*Pelomonas* sp. K89 (NCBI Acc. #KU233240), source: drinking water sand filter, Belgium	99.1

^1^ https://www.ncbi.nlm.nih.gov/nuccore/, accessed on 20 February 2022.

## Data Availability

Sequences of 16S/18S rDNA fragment of the new bacterial and fungal isolates determined under the study are deposited in NCBI GenBank Acc. # MT192546.1; MT192545.1; MT192544.1; MW316031.1; MW316054.1; MW316043; MW341486.1; MW341467.1; MW341490.1; MW341491.1. The pure bacterial cultures are deposited in Registration No. in All-Russian Collection of Industrial Microorganisms and are available under registration numbers VKPM B 13662; VKPM B 13661 and VKPM B 13660.

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
