# Peer review of "Heterotrophic Microbiota from the Oligotrophic Waters of Lake Vostok, Antarctica"

_ijerph, 2022, doi:10.3390/ijerph19074025_

Round 1
Reviewer 1 Report
The article is well written, the objectives of the work are clear, as well as its conduct.
I felt a lack of information on the DNA polymerases used to carry out the amplifications, to know if they are low or high fidelity.
Otherwise, I have no corrections or suggestions to make, and I consider the article suitable for publication
Author Response
Уважаемый рецензент,
спасибо за критику. Мы уточнили название термостабильной полимеразы, используемой для ПЦР-клонирования фрагментов рДНК, используемых для молекулярной идентификации микроорганизмов, следующим образом: «Q5 ® Hot Start High-Fidelity DNA Polymerase (New England Biolabs, США) с добавленным производителем буфером. используется для ПЦР».
Reviewer 2 Report
Review for IJERPH- 1629907
Dear Authors,
This is an interesting paper regarding microbiota from Lake Vostok but it needs to be improved in some aspects. You can find below the specific comments to help you in the revision and correction. Specially, results and discussion sections should be improved, although the second part of the discussion (from line 315) is good and more the way you should follow.
Lines 32-47: Is there any reference about facts and history milestones for this paragraph?
Line 77: Instead of begin with the number of the cite indicate authors and year, then put the “in-brackets cite” and continue the sentence. (The same for line 270, 298, 299, 300, 305…)
Line 85: Maybe to include a brief information just to say where the Lake Fryxel is located. (The same for next line and lakes) Also, consider to include some picture/photograph of the area that could improve the section.
Lines 87-99: All this information needs to refer, again, to reference 11, if it is enough to cover that information, or to add another one.
Line 113-115: It would be better just to indicate once the LB composition and then say that Petri dishes were prepared with 1x, 0.5x and 0.1x of LB, instead of repeat the LB composition three times
Line 126: Specify the medium used for the passages.
Line 141-151: Consider to put this information onto a table to be clearer
Line 170: cells were mainly lysed by this step, not only homogenized.
Line 174-178: How did you obtain the sequence of PCR product? Indicate it in the manuscript. You should mention, at least, that a PCR was performed, before to talk about a PCR product.
Line 184-188: From specimens or from raw samples collected from Lake?? I understand that specimens appeared after culturing the samples collected in the different plates, and then you did the characterization. Check it, please.
Line 192-196: These numbers of colonies, which concentration of LB-plates refers to?
Figure 1. Please, specify which LB concentration is the plate of the picture
Line 208: Plating…original samples or colonies from LB-plates?
Table1. In the column regarding cell morphology the two last species do not indicate cell morphology information but just the microbial group based on temperature growth/environment.
Line 221-222: Re-plated in which medium?
Line 241-242: This statement should be in the discussion section not in results.
Line 246-252: Just my opinion: I do not find necessary to indicate the whole phylogeny of each species.
In view of the description of these results, I would suggest to consider, if it is possible, to include pictures of the different types of colonies described, not only the sole example include.
264-277: All this information is introduction and should not be in discussion section. Moreover, is quite repetitive with information already given.
Line 290: This paragraph requires several references for the information given about microbiota of Lakes mentioned.
Lines 291-297: What do you want to mean by pointing out by this paragraph? The idea is not clear neither are the ideas linked by the sentences.
Line 306-312 are methods, this should not be here. The same for Table 2 that are results.
Line 341: Described elsewhere “in” where?? This paper? Another one? Which one?
Line 354: it needs a reference. The same for the next 5 lines of information.
Line 358-359: Which signs are exhibited?
Reviewer 3 Report
A report for: ijerph-1629907 Heterotrophic Microbiota from the Oligotrophic Waters of Lake Vostok, Antarctica
I have reviewed the manuscript that focused on the characterisation of the species able to grow on complete organic media in the presence of the oxygen under mesophilic conditions. The manuscript, that contains new data, is primarily a data report with limited hypothesis testing and scientific interpretation of the results So, it is difficult to recommend this manuscript for publication in the present form. In my opinion, the manuscript needs some work to become suitable for publication.
- Abstract should be started with a couple of sentences to show the background and why this research needed to be conducted.
- The introduction is not cohesive. It is just a compilation of information. The justification for objects selection is needed. Clarify the objectives explicitly.
-A map or GPS coordinates, or a table describing the sites might help here.
-The sampling design described here is confused.
-Line 32-47 it is neccessary?. In all case please reduce the extension.
-It is necessary and advisable to expose some final conclusions.
-I susggest to include some photos of the área (if it is possible).
I wish those changes will contribute to improve your paper.
Reviewer 4 Report
Epova et al. interesting work regarding bioprospecting of Lake Vostok, the deepest lake of Antarctica states the isolation and description of strains belonging to eight heterotrophic microbial species ,six bacterial strains belonging to the species Microbacterium testaceum and
trichothecenolyticum, Brevundimonas diminuta, Sphingomonas oligophenolica, Sphingomonas sp. and Sphingobium limneticum; and two fungal strains belonging to Dendryphion sp. and Cladosporium
fusiforme. The article requires a general reformulation as in making things simpler and easier for the reader, a full english revision should be performed as well.
2. Materials and Methods
Sample Collection and Analysis, Enrichment and Isolation
This subchapter is well written.
Morphology
State the method in a step by step manner
Molecular Study of Uncultured Microbiota; Molecular Phylogenetic Analyses are well written
Conclusion section is missing and needs to be added as it is mandatory, separate the discussion chapter from conclusions and clearly state them
Round 2
Reviewer 2 Report
Dear authors,
The manuscript has improved but still it needs some revision and amendments or clarifications. I have left in red my comments to the responses the have to be adressed. (Also, missed from the comments-document: in line 393 correct dominance instead of dominonce)

Reviewer 3 Report
The authors detail all the changes that I suggested to improve the manuscript. The list of points for to be consider by the authors has been answer. Then, it can be accepted.
Author Response
Dear reviewer, please see the new version of the article that we posted